# Drug susceptibility profiles and factors associated with non-tuberculous mycobacteria species circulating among patients diagnosed with pulmonary tuberculosis in Tanzania

**Togolani Godfrey Maya** [1] *, **Erick Vitus Komba**[1], **Gloria Ivy Mensah**[2], **Peter Masunga Mbelele**[3], **Stella George Mpagama**[3], **Sayoki Godfrey Mfinanga**[4], **Kennedy Kwasi Addo**[2], **Rudovick Reuben Kazwala**[1]

1 Department of Veterinary Medicine and Public Health, College of Veterinary Medicine and Biomedical Sciences, Sokoine University of Agriculture, Morogoro, Tanzania, 2 Department of Bacteriology, Noguchi Memorial Institute for Medical Research, University of Ghana, Accra, Ghana, 3 Kibong'oto Infectious Diseases Hospital, Moshi, Tanzania, 4 Muhimbili Research Centre, National Institute for Medical Research, Dar es Salaam, Tanzania

* tmaya.tz@gmail.com

## Abstract

### Background

While most Non-tuberculous mycobacteria (NTM) are saprophytic, several species have been associated with human diseases, from localized infection to disseminated diseases. Pulmonary NTM infections lead to TB-like disease called NTM pulmonary disease (NTM-PD). Due to variation in treatment options among NTM species, it is necessary to identify the species and determine drug susceptibility profiles to inform the choice of appropriate regimen for the disease.

### Design

A total of 188 culture-positive isolates from patients diagnosed with TB were screened for NTM at the Central Tuberculosis Reference Laboratory. All NTM were further speciated using GenoType® Mycobacterium—Common Mycobacterium and Additional species (GenoType® CM/AS) kit. *Mycobacteria avium* complex (MAC) and *Mycobacteria abscessus* complex (MABC) which could not be identified with the test to species were subjected to GenoType® Mycobacteria NTM-DR for further speciation. Using the same test, identified MAC and MABC were genotyped to determine the drug susceptibility profile for each isolate to macrolide and aminoglycosides.

### Results

Of all isolates identified as mycobacteria, 24 (13%) were NTM. Fifteen isolates could be identified to species level of which prevalent species was *M. avium* sub. *intracellulare* 4

**Data Availability Statement:** All relevant data are within the paper and its Supporting Information files.

**Funding:** The research for this paper was carried out within the framework of the DELTAS Africa Initiative [Afrique One-ASPIRE /DEL-15-008]. Afrique One-ASPIRE is funded by a consortium of donor including the African Academy of Sciences (AAS) Alliance for Accelerating Excellence in Science in Africa (AESA), the New Partnership for Africa's Development Planning and Coordinating (NEPAD) Agency, the Wellcome Trust [107753/A/15/Z] and the UK government. The funders had no role in study design, data collection and analysis, decision to publish, or preparation of the manuscript.

**Competing interests:** The authors have declared that no competing interests exist.

(27%). A total of 10 isolates were MAC (n = 6) and MABC (n = 4) were subjected to Geno-Type® Mycobacteria NTM-DR for determination of macrolide and aminoglycoside susceptibility. Three of the four MABC had a mutation at the T28 position of the *erm (41)*. All MAC were susceptible to both drugs.

## Conclusion

In this study, MAC was the most frequently isolated NTM species followed by MABC. While all MAC and MABC identified, were susceptible to aminoglycosides, three MABC were resistant to the macrolides due to mutation at position 28 of the *erm (41)* gene. For this, it is important for clinicians need to rule out NTM, understand species and their drug susceptibility for optimal case management.

## Background

Non-tuberculous mycobacteria pulmonary disease (NTM-PD) results from lung infection of atypical mycobacteria. These bacteria have also been known as Mycobacteria other than tuberculosis (MOTT) or environmental mycobacteria. On the other hand, *Mycobacteria tuberculosis* complex species (MTBC) cause Tuberculosis while *Mycobacterium leprae* cause leprosy. This disease presents with similar features to TB and has frequently been referred to as atypical TB [1, 2]. A study in 2013 reported a total of 140 NTM species [3] in 2019 but there are more than 200 species [4]. While some NTM species cause disease especially in people with other underlying diseases, most of them are saprophytic [5].

More than 30 NTM spp. have been found to have clinical implications on human health [6–9]. These can further be categorized into two groups: slow-growing and rapid growing [10]. In Tanzania, a case that was misdiagnosed as MDR-TB after relapse and failure of first-line TB drugs regimen was later discovered to be a result of *M. yongonense* infection, one of the novel NTM [11]. Studies carried out in Kilimanjaro and Arusha discovered NTM causing septicemia and lymphadenitis [12, 13].

There is currently no clear evidence of human-to-human transmission of NTM-PD documented [7, 14]. A multicenter NTM prevalence study among people with CF indicated no strong enough strain similarity to prove human to human transmission [15]. In general, MAC are among the most isolated spp. of NTM; more specifically *M. avium*, *M. intracellular*, and *M. kansasii* [16]. A systematic review and meta-analysis of 37 articles on NTM in the Southern Saharan region revealed a prevalence of 7.5% of pulmonary NTM with MAC prevailing by 28% and in 19 of the studies [17]. A study similar to the current which was done on culture-positive samples from a National TB prevalence survey revealed an NTM proportion of 54.3% with *M. fortuitum* being the most isolated species [18].

The proportion of cases of NTM is significantly high among TB presumptive cases in Tanzania. Two studies that were carried out in Tanga indicated 9.7% and 8.1% of presumptive TB patients had NTM. *M. interjectum* isolates were found the leading NTM in the catchment (16.7%) followed by *M. intracelallare* 11.1%. The prevalence was high among patients with other underlying diseases like AIDS and Cystic Fibrosis (CF) [19, 20]. Non-tuberculous mycobacteria diversity has also been investigated from human-animal interaction in the Serengeti ecosystem where 36.7% were *M. intracellulare* [21].

Success in the management of NTM is largely limited by a lack of knowledge on species circulating and their drug susceptibility profiles. This is because the treatment regimen is species-dependent and most of the clinically relevant NTM species are naturally resistant to conventional TB drugs [22, 23]. While some clinically relevant NTM species can respond to conventional TB drugs such as *M. kansasii*, the rest are naturally resistant to these drugs [24]. This poses a high risk of unsuccessful treatment outcomes and long hospitalization of NTM cases misdiagnosed as TB.

The current study sought to identify species and drug susceptibility profiles of non-tuberculous mycobacteria commonly isolated at the Central Tuberculosis Reference Laboratory (CTRL) in Tanzania.

## Material and methods

### Study site

Testing of TB in Tanzania is organized into different levels of laboratories from National/central, zonal, regional, district, health center, and dispensary level. However, this study was conducted at the Central Tuberculosis Reference Laboratory (CTRL) in Dar es Salaam. In addition to receiving culture-positive isolates from the zonal laboratories for comprehensive routine surveillance of drug resistance, CTRL serves as zonal in its catchment, Eastern zone (Dar es Salaam, Pwani, Morogoro, Lindi, and Mtwara). Others include; Kibong'oto Hospital Laboratory (Northern zone), Pemba Public Health Laboratory (Pemba and Unguja), Bugando Medical Centre Laboratory (Lake Zone), Mbeya Referral hospital laboratory (Southern highlands), and Dodoma Regional Referral Hospital laboratory (Central zone).

In implementing universal drug susceptibility testing (DST) coverage strategy, Tanzania implements a culture of sputum specimens from bacteriologically confirmed TB patients for comprehensive conventional DST. For 2018 a total of 27,201 pulmonary TB cases were bacteriologically confirmed in Tanzania for which drug susceptibility status to at least one drug was required [25]. Samples are received at CTRL for various reasons including DR-TB surveillance (by Xpert MTB/Rif, Hain line probe assay and/or conventional DST), TB diagnosis (from clinically diagnosed patients), and treatment monitoring. A total of 2624 sputa and 538 isolates were received at CTRL in 2018 for culture and DST respectively [26].

### Study design and sampling

This was a cross-sectional study in which isolates from culture-positive sputa specimens were conveniently selected and screened for NTM. Identified NTM were speciated and tested for genotypic drug susceptibility. Only pulmonary TB patients who were bacteriologically confirmed and their isolates available were included in this study. On the other hand, pulmonary TB patients who turned culture-negative and those whose isolates could not be found were excluded from this study. A priority was given to isolates that grew on a culture medium containing para-aminobenzoic acid (PNB) during conventional DST as these were the most probable NTMs. The isolates were genotyped using GenoType® Mycobacterium, Common Mycobacteria, and Additional species (GenoType® CM/AS) for NTM species identification, and then the species were genotyped using GenoType® Mycobacteria NTM-DR for drug susceptibility to the recommended drugs. CTRL culture registers and the list of culture-positive isolates received between November 2019 and August 2020 provided the sampling frame from which samples were conveniently selected. Each of the selected positive cultures (mycobacteria isolate) from primary culture media were transferred to a 2ml cryotube containing 15% glycerol in Tryptic Soya Broth (TSB). These isolates were then stored at– 200C until used.

This study only included isolates received at CTRL for diagnosis, treatment monitoring, and conventional TB DST. While most of the isolates were from newly diagnosed patients before treatment initiation, few were isolates from patients under treatment monitoring (follow up). This is because NTM, especially MAC, has been frequently isolated from patients under TB treatment [26]. Based on the formula below at least 134 AFB culture positive isolates were supposed to be analyzed for this study.

Where;

N = Sample size t = 1.96 L = 0.05 P = 0.097

T = student t–value

L = Significance level

SD = Standard Deviation

$$n \geq 134$$

## Data collection

**Patients information (from patients' test request/report forms).** Patients' demographic characteristics (age, gender, district, region, etc.) and clinical information such as HIV status, the diagnostic test used, initial local GeneXpert or AFB smear, and culture results were collected. Laboratory test request form and TB LIS (electronic laboratory information system) provided a source of patients' data.

**Extraction and storage of DNA.** For genotyping, DNA were extracted chemically using GenoLyse® (Hain Lifescience, Nehren, Germany) protocol [27]. In case DNA could not be GenoTyped® on the same day of extraction, it was stored in a deep freezer at– 20˚C. After genotyping, the DNA was transferred to– 80˚C for long-term storage.

**Species identification.** PCR was carried out in the GTQ Cycler 96 using the protocol in the user guide (Hain Lifescience, Nehren, Germany). Speciation was carried out using Geno-Type®CM/AS (Hain Lifescience, Nehren, Germany) protocol on TwinCubator (Hain Lifescience, Nehren, Germany). Species-specific probes are mounted to the DNA strips to determine complementary strands from amplified DNA samples [28, 29]. **Table 1** below shows that MAC and *Mycobacterium abscessus* complex (MABC) that could not be speciated by this technique were further speciated using GenoType® NTM-DR (Hain Lifescience, Nehren, Germany) protocol [30]. DNA amplicon that could not be identified to species level the same day of amplification were stored at– 200C. On the other hand, isolates that GenoType® CM/AS assay could not GenoType® to species level are stored at– 800C for sequencing of 16S rRNA gene.

**Drug susceptibility testing of NTM.** Molecular drug susceptibility profiles of *Mycobacteria avium* Complex and *M. abscessus* complex isolates were carried out using the GenoType® NTM-DR. This PCR technique involves amplification of the *erm (41)* (erythromycin ribosomal methylase) gene which encodes for 23S peptidyl transferase of the large (50S) subunit of rRNA. GenoType® NTM-DR detects mutation that may result in the development of resistance to macrolides. However, *erm (41)* is available only in MABC with exception of a few spp. Moreover, this also involves amplification of the *rrs* gene. Mutation(s) in the *rrs* gene which encodes for 16S rRNA results in resistance to aminoglycosides [31].

## Quality control

For each investigation, known positive control strains *M. fortuitum* (ATCC® 6841™) and *M. kansasii* (ATCC® 12478™) were included. On other hand, DNAse-free water (molecular grade) was used as a negative control for quality assurance. The machines used for testing of

**Table 1. Banding pattern and presumed species.**

| Band pattern CM | Frequency | Band pattern AS | Band pattern NTM-DR | Presumed species |
|---|---|---|---|---|
| 1,2 | 9 | Not Applicable | Not Applicable | Negative for Mycobacteria |
| 1,2,3,10,16 | 155 | Not Applicable | Not Applicable | MTBC |
| 1,2,3,5,6,10 | 3 | Not Applicable | SP4, SP5, SP6, SP9, SP10 | *M. abscessus sub. abscessus* |
| 1,2,3,5,6,10 | 1 | Not Applicable | SP5, SP6, SP7, SP9, SP10 | *M. abscessus sub. bolletti* |
| 1,2,3,9 | 4 | Not Applicable | SP2 | *M. avium sub. intracelluare* |
| 1,2,3,7,14 | 2 | Not Applicable | Not Applicable | *M. fortuitum* group |
| 1,2,3,10,11 | 1 | 1,2,3,4,6 | Not Applicable | *M. simiae* |
| 1,2,3,6,10,11 | 1 | 1,2,3 | Not Applicable | *M. szulgai* |
| 1,2,3,10 | 4 | 1,2,3 | Not Applicable | Indeterminate* |
| 1,2,3,10 | 3 | 1,2,3,12 | Not Applicable | Indeterminate* |
| 1,2,3,5,6,10,16 | 1 | 1,2,3,12 | Not Applicable | (Mixed MTBC and NTM) |
| 1,2,3 | 1 | 1,2,3 | Not Applicable | Indeterminate* |
| 1,2,3,4 | 2 | Not Applicable | SP1 | *M. avium* |
| 1,2,3,10,12 | 1 | Not Applicable | Not Applicable | *M. kansasii* |

* Need targeted sequencing of 16S rRNA gene to identify species

samples had passed method validation that ensures they produce reproducible results and match manufacturer specifications.

## Data management, confidentiality, and statistical analysis

Patients' descriptive data and genotyping analytical data were collected and cleaned in Microsoft Excel® 2010 and then imported to R software for analysis. These include independent variables such as age, gender, NTM species, HIV status, patients' category, and reason for testing (diagnosis or treatment monitoring). Dependent (response) variables include; NTM species and drug susceptibility (i.e. susceptible or resistant).

## Ethical clearance

This study protocol was approved by the Sokoine University of Agriculture Institutional Research Review Board and ethical clearance was provided by the Institutional Research Ethical Committee of the National Institute for Medical Research (NIMR) (NIMR/HQ/R.8a/Vol. IX/3245). The need for participants' consent for this study was waived by the ethical committee. Study Identification numbers, and laboratory serial numbers were used instead of patients' names so the identity of each participant remained fully anonymous. Data obtained was kept in a computer having its access protected using a password and hard copies were kept in a lockable draw.

## Study results

### Patients' demographic characteristics

Specimen analyzed in this study came from 188 patients diagnosed with pulmonary TB of whom 45 (24%) were females. Overall mean age was 43 (STD: 15) years with females being 38 and males 48. The 35 to 44 years age group had the largest number (30%) of study subjects. The mean age of patients with NTM was 47 (STD: 15). **Fig 1** indicates study subjects' distribution by age groups. The number of patients who had a known HIV status was 144 of whom 34

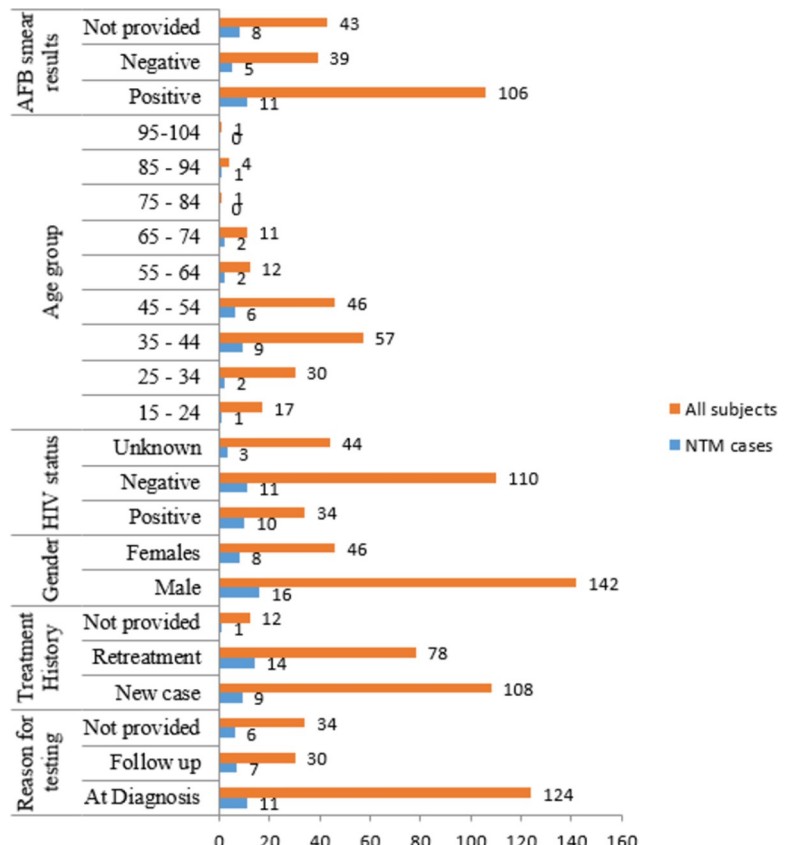

**Fig 1. Variation of the number of non-tuberculous mycobacteria between various independent variables.**

(24%) were positive for the virus. While of 110 who were HIV-negative, 11 (10%) had NTM, of the 34 patients with HIV, 10 (29%) had NTM infection. The remaining three isolates had their patients with unknown HIV statuses. About treatment history, a total of 176 patients records were available of which 113 (64%) were new TB patients while 63 (36%) had previously been treated for the disease.

## Non-tuberculous mycobacteria species and drug susceptibility

Isolates analyzed in this study resulted from culture-positive sputum from patients diagnosed with pulmonary TB. Of the 188 isolates tested, 179 (95%) were positive for the genus mycobacteria while the remaining were negative. Among the positive samples, 24 (13%) were NTM while the rest were MTBC. GenoType® Mycobacterium CM/AS and GenoType® Mycobacterium NTM-DR could identify 15 (63%) NTMs to their species levels. This study revealed *Mycobacterium Avium* Complex (MAC) as the most prevalent group. There were six (25%) species belonging to this group (four *M. intracellulare* and two *M. avium)*. On the other hand, there were four (17%) *Mycobacterium abscessus* complex (MABC), three *M. abscessus sub. abscessus*, *one M. bollettii*, and two (8%) in the Fortuitum group. Other species included one (4%) *M. kansasii*, one (4%), *M. simiae*, and one (4%) *M. szulgai*.

All MAC and MABC (10 isolates) were subjected to GenoType® NTM-DR for susceptibility testing to macrolides (azithromycin and capreomycin) and aminoglycosides (kanamycin, amikacin, and gentamycin). While MABC resistance to macrolides results from mutation conferred in either or both of *rrl* and *erm (41)* genes, MAC resistance to this drug results from a

**Table 2. Summary wild type bands and associated mutations in the GenoType® Mycobacterium NTM-DR assay.**

| MAC/MABC Species | | Gene | | | | | | Macrolides | Aminoglycosides | Frequency |
|---|---|---|---|---|---|---|---|---|---|---|
| | | *erm (41)* | | *rrl* | | *rrs* | | | | |
| | | C28 | T28 | WT | MUT | WT | MUT | | | |
| | Intracellulaire | NA | NA | + | - | + | - | S | S | 4 |
| | Abscessus | - | + | + | - | + | - | R | S | 2 |
| | Abscessus | + | - | + | - | + | - | S | S | 1 |
| | Bollettii | - | + | + | - | + | - | R | S | 1 |
| | Avium | NA | NA | + | - | + | - | S | S | 2 |

WT–Wild type, MUT–Mutation, S–Susceptible, R–Resistant and

NA–Not applicable

mutation in the *rrl* gene only. Of the 10 NTMs tested for drug susceptibility, 3 were found to have a mutation (T28 point mutation) in the *erm (41)* of which Cytosine is replaced by Thymine at position 28 of the gene. However, there was no macrolide resistance mutation detected in the *rrl* gene. Likewise, no mutation was detected for aminoglycosides resistance (in the *rrs* gene) for both MABC and MAC (**Table 2**).

## Factors associated with non-tuberculous mycobacterial infection

**Fig 2** illustrates NTM proportions among levels of some important variables investigated in this study. The prevalence of NTM was higher among people who were previously treated (category) for TB 14 (58%) compared to new patients (p = 0.0656). Most cases of NTM were in the reproductive age between 35 and 44 years which is similar to TB. There was a significant relationship between gender and NTM infection where males were more likely to be positive than females (p = 0.008655). While 11(46%) tested positive for NTM among patients for specimen collected just before treatment initiation, 6 (25%) patients tested positive for NTM after some months of treatment (p = 0.004426) suggesting that more NTM are recovered at diagnosis than during treatment monitoring. NTM were recovered in 9 (26.5%) patients among those who were positive for HIV (n = 34, 18%). This study could not find a significant relationship between the type of mycobacterial infection (MTBC or NTM) and HIV status (p = 0.1708).

## Discussion

Increasing numbers and burden of NTM among presumptive TB cases makes it of paramount importance for National TB Programs (NTP) to consider screening for NTMs from among TB presumptive cases especially for patients at risk [32, 33]. Furthermore, knowledge of prevailing species and their drug susceptibility pattern is critical for correct diagnosis and selection of appropriate drug regimens [6].

The current study aimed at determining NTM species circulating among patients diagnosed with TB and their drug susceptibility profiles. Drug susceptibility profiles were specifically performed to MAC and MABC on the recommended drugs. There have been several reports of NTM among patients on TB treatment in Tanzania [11, 34]. In this particular study, the interest was in characterizing NTM often recovered from sputum of presumed pulmonary TB patients received at CTRL for species and molecular drug susceptibility tests.

In this study, *M. intracellulare* (MAC) was the most prevalent species as has been observed in previous studies [6, 16, 35]. This was followed by MABC species which have also been well documented in the causation of NTM-PD. Moreover recovered *M. kansasii* has commonly been associated with pulmonary disease. The fact that coinfection occurs between MTBC and

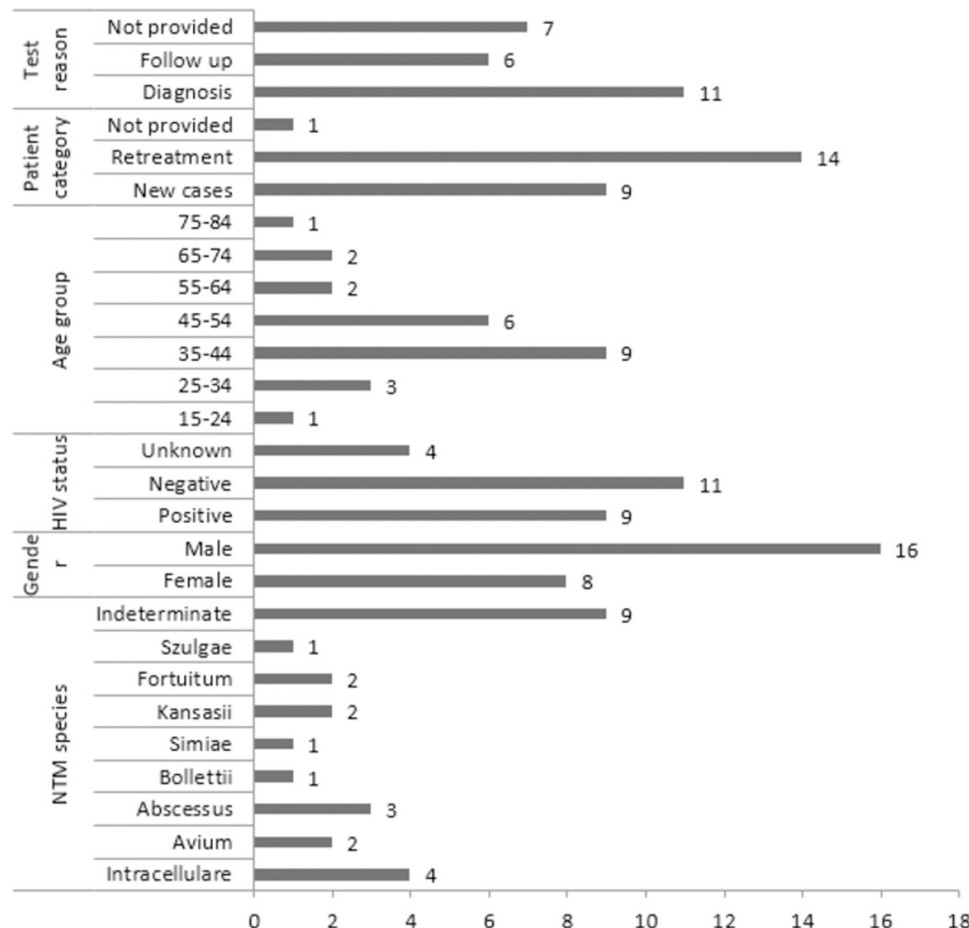

**Fig 2. Factors associated with Non-tuberculous mycobacteria among patients with pulmonary TB.**

NTM was also complemented in our findings where one of the isolates had both of these mycobacteria strains [36]. For this, the use of rapid chromatographic assays to screen- out NTM among mycobacteria culture-positive, and exclusion of NTM in laboratory reporting in Tanzania blinds the existence of these pathogenic NTM among the patients. To remedy the situation, it will be advisable to report culture-positive NTM among patients with chronic or acute pulmonary disease. This study is the first to recover *Mycobacterium abscessus* sub. *bolletii and M. szulgae* isolates in Tanzania [12, 20, 37, 38]. The greater number of species identified (eight different species) indicates the diversity of NTM species circulating among patients treated for TB in Tanzania. A national NTM prevalence survey is warranted to enable the estimation of the burden and treatment options in Tanzania.

In this study, NTM were more frequently isolated from males (twice) when compared to females. This finding is similar to what has been reported by González et al. [39] but contrary to this study, Park et al., and Dodd et al [40, 41] reported that NTM was isolated from more females than males. Findings from this study are similar for Tuberculosis on gender where more males contract the disease than females with the reason being a higher number of daily close and casual contacts in men [41]. NTM distribution by age group is also similar to the distribution of TB where more patients are within the ages of 35 to 44 years. We observed a proportional increase with age where patients with higher age were more likely to have NTM-PD [42].

Our results support the fact that patients previously treated for TB (relapse, loss to follow-up, treatment failure, and MDR contact) have higher chances of being positive for NTM infection than new cases [43]. More epidemiological studies are required to investigate the factors that favor NTM colonization and infection in this group of patients. While most of the recovered NTM were from samples collected at the time of TB diagnosis, some were identified from samples of patients after several months of TB drug administration. On the other hand, despite the high number of NTMs among HIV infected compared to non-infected, the current study could not support the hypothesis that people with HIV have higher chances of NTM infection in comparison to HIV-negative, a similar observation to the one made earlier [44]. This might have resulted from differences in the sample size and sampling methods employed.

Erythromycin resistance methylase gene (*erm (41)*) exists in MABC species like *M. abscessus* sub. *abscessus* and *M. abscessus* sub. *bollettii* [35]. Other species in this group like *M. massilience* lack the target base at position 28 which results from a large deletion of a 397 base-pair sequence in which this target region is included. Such species have a nonfunctional *erm (41)* gene thus do not express resistance to macrolides. Mutation at nucleotide position 28 of this gene where Cytosine is replaced by Thymine results in phenotypic drug resistance of these two species to macrolides [45, 46]. Two of the three identified *M. abscessus* sub. *abscessus* and one *M. bollettii* were detected with this mutation indicating prevailing resistance among the MABC population. However, these strains were all susceptible to the aminoglycosides. Management of MABC that are resistant to macrolide can follow other recommended guidelines [22]. On the other hand, no mutation was detected in both *rrl* and *rrs* genes in this study. This indicates the susceptibility of the remaining identified MABC and MAC to both drugs. A study carried out in Ghana on National prevalence survey samples also revealed susceptibility of all isolated MAC and MABC to aminoglycosides [18].

## Study limitations

While most of the isolates characterized in this study came from patients within their first month of TB treatment, few were from patients after some months of treatment. There is no evidence of these isolates to meet diagnostic criteria for non-tuberculous mycobacteria causation of lung disease as described in the ATS/IDSA Statement (Griffith *et al*., 2007) and the British Thoracic Society Guideline for the management of non-tuberculous mycobacterial pulmonary disease (NTM-PD) (Haworth *et al*., 2017). Moreover, the genotyping techniques employed in this study could not identify all isolated NTM to species level; nine isolates were left un-speciated. Aminoglycosides and macrolides are the recommended important components treatment of NTM-PD however many other drugs can be combined in developing an efficient regimen.

## Conclusion

This study recovered a total of eight different NTM species from sputum of pulmonary TB patients submitted for routine culture and drug susceptibility testing. *Mycobacteria avium* complex was the most frequent, (n = 6) isolate followed by MABC (n = 4). Both of these require medical attention when diagnosed as they have been associated with pulmonary disease. On the other hand, while all MAC and MABC were susceptible to aminoglycosides, three MABC had a mutation that causes resistance to macrolides.

## Supporting information

**S1 File.**
(XLSX)

## Acknowledgments

We are grateful to the management of the Central Tuberculosis Reference Laboratory (CTRL) in Tanzania for accepting and giving permission on the use of the archived samples. Moreover, to the laboratory staff for support, they gave during retrieving the archived samples, DNA extraction, and genotyping. Also, we acknowledge the CTRL data management team who assisted in accessing the patients' data and in analysis. Lastly but not least we acknowledge the TB patients whose samples were used in this analysis.

## Author Contributions

**Conceptualization:** Togolani Godfrey Maya, Erick Vitus Komba, Gloria Ivy Mensah, Peter Masunga Mbelele, Stella George Mpagama, Rudovick Reuben Kazwala.

**Data curation:** Togolani Godfrey Maya, Erick Vitus Komba.

**Formal analysis:** Togolani Godfrey Maya, Erick Vitus Komba.

**Funding acquisition:** Togolani Godfrey Maya, Gloria Ivy Mensah, Sayoki Godfrey Mfinanga, Kennedy Kwasi Addo, Rudovick Reuben Kazwala.

**Investigation:** Togolani Godfrey Maya, Erick Vitus Komba, Gloria Ivy Mensah.

**Methodology:** Togolani Godfrey Maya, Erick Vitus Komba, Gloria Ivy Mensah.

**Project administration:** Togolani Godfrey Maya, Erick Vitus Komba, Gloria Ivy Mensah, Sayoki Godfrey Mfinanga, Kennedy Kwasi Addo.

**Resources:** Togolani Godfrey Maya, Gloria Ivy Mensah, Sayoki Godfrey Mfinanga, Kennedy Kwasi Addo, Rudovick Reuben Kazwala.

**Software:** Togolani Godfrey Maya.

**Supervision:** Erick Vitus Komba, Gloria Ivy Mensah, Sayoki Godfrey Mfinanga, Kennedy Kwasi Addo, Rudovick Reuben Kazwala.

**Validation:** Togolani Godfrey Maya, Erick Vitus Komba, Gloria Ivy Mensah.

**Visualization:** Togolani Godfrey Maya, Erick Vitus Komba, Gloria Ivy Mensah.

**Writing – original draft:** Togolani Godfrey Maya, Erick Vitus Komba, Gloria Ivy Mensah.

**Writing – review & editing:** Togolani Godfrey Maya, Erick Vitus Komba, Gloria Ivy Mensah, Rudovick Reuben Kazwala.

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
