## [Decision Letter · Decision Letter 0]

10 Dec 2021

PONE-D-21-34246Drug susceptibility profiles of non-tuberculous mycobacteria species circulating among patients diagnosed with pulmonary tuberculosis in TanzaniaPLOS ONE

Dear Dr. Maya,

Thank you for submitting your manuscript to PLOS ONE. After careful consideration, we feel that it has merit but does not fully meet PLOS ONE’s publication criteria as it currently stands. Therefore, we invite you to submit a revised version of the manuscript that addresses the points raised during the review process.

 Please submit your revised manuscript by Jan 24 2022 11:59PM. If you will need more time than this to complete your revisions, please reply to this message or contact the journal office at plosone@plos.org. Please include the following items when submitting your revised manuscript:A rebuttal letter that responds to each point raised by the academic editor and reviewer(s). You should upload this letter as a separate file labeled 'Response to Reviewers'.A marked-up copy of your manuscript that highlights changes made to the original version. You should upload this as a separate file labeled 'Revised Manuscript with Track Changes'.An unmarked version of your revised paper without tracked changes. You should upload this as a separate file labeled 'Manuscript'.

We look forward to receiving your revised manuscript.

Kind regards,

Seyed Ehtesham Hasnain

Academic Editor

PLOS ONE

Journal Requirements:

The research for this paper was carried out within the framework of the DELTAS Africa Initiative [Afrique One-ASPIRE /DEL-15-008]. Afrique One-ASPIRE is funded by a consortium of donor including the African Academy of Sciences (AAS) Alliance for Accelerating Excellence in Science in Africa (AESA), the New Partnership for Africa’s Development Planning and Coordinating (NEPAD) Agency, the Wellcome Trust [107753/A/15/Z] and the UK government.

The research for this paper was carried out within the framework of the DELTAS Africa Initiative [Afrique One-ASPIRE /DEL-15-008]. Afrique One-ASPIRE is funded by a consortium of donor including the African Academy of Sciences (AAS) Alliance for Accelerating Excellence in Science in Africa (AESA), the New Partnership for Africa’s Development Planning and Coordinating (NEPAD) Agency, the Wellcome Trust [107753/A/15/Z] and the UK government.

Additional Editor Comments:

Major Revision

Reviewers' comments:

Reviewer's Responses to Questions

**Comments to the Author**

1. Is the manuscript technically sound, and do the data support the conclusions?

Reviewer #1: Yes

Reviewer #2: No

2. Has the statistical analysis been performed appropriately and rigorously? 

Reviewer #1: I Don't Know

Reviewer #2: I Don't Know

3. Have the authors made all data underlying the findings in their manuscript fully available?

Reviewer #1: No

Reviewer #2: No

4. Is the manuscript presented in an intelligible fashion and written in standard English?

Reviewer #1: Yes

Reviewer #2: No

5. Review Comments to the Author

Reviewer #1: In this study authors have tried to identify Mycobacterium species in NTM-PD cases so that better treatment option can be done. There are a few concerns, under following headings

1. Study sites

In 2018 a total of 27 201 TB cases were bacteriologically confirmed in (26)- Incomplete sentence.

2. Quality control

Is it DNA free water or DNAse free water?

3. Factors associated with Non-tuberculous mycobacteria infection among patients with pulmonary TB

Similarly, this study found an association between NTM infection and age of patients (p

270 = 0.07367).- Elaborate

4. Results

Table 1 is not clear,, not making any sense or give better explanation in results section

5. Discussion

Similarly to the González, SG (39) and contrary to Park SC and Dodd PJ (41,42) NTM were more frequently isolated from males when compared to females- Rephrase

6. Conclusion

followed up by MABC (n=4); remove “up”

Reviewer #2: Overall assessment: Major revision

The paper entitled " Drug susceptibility profiles of non-tuberculous mycobacteria species circulating among patients diagnosed with pulmonary tuberculosis in Tanzania" by Togolani Godfrey Maya et al., have tried to address an important issue of drug susceptibility associated with NTMs diagnosed with Pulmonary TB. It is well documented that many NTM) are saprophytic, however several species of NTM have been shown to cause human diseases, Pulmonary NTM infectious lead to TB-like disease called NTM pulmonary disease (NTM-PD). Due to variation in treatment options among NTM species, it is necessary to identify the species and determine drug susceptibility profiles to inform choice of appropriate regimen for the disease. 188 isolates from TB patients diagnosed were screened for NTM and were genotyped to determine drug susceptibility profile to macrolide and aminoglycosides. Data suggest that out of all the isolates 24 were NTM of which 15 were M. avium sub. intracellulare 4 and a total of 10 isolates were MAC (n=6) and MABC (n=4). Further authors have shown that the MAC were the most frequently isolated NTM species followed by MABC. While all MAC and MABC identified, were susceptible to aminoglycosides. Although the rationale of this study is interesting but it should be rejected because of lacks depth, clarity and possible explanation for mechanists, poor data presentation and

Comments:

1. Sample size in the study for NTM is too small. Majority of the NTM reported is in the HIV + cases (29%) as compared to HIV negative cases (10%). What could be the possible explanation.

2. It would be better to include patients with no history of HIV infection. What about other immune comprised conditions. Dose these NTMs are also seen in similar way.

3. No proper explanation of inclusion and exclusion criterion for including Pulmonary TB patients

4. In abstract it is written 188 TB patients; however, data shown is for 144 HIV co-infected patients. Where 44 patients had no history of HIV????, if so, how was the status of NTM in these patients. There is no clarity in this regard???

5. To do justice with title of the manuscript, authors should include other TB conditions or change it as TB patients having HIV history.

6. Figures are poorly represented.

7. Why M avium was not used for drug susceptibility profile test in Table 2?

8. Grammatical checks need to be done throughout the manuscript.

6. PLOS authors have the option to publish the peer review history of their article (what does this mean?). If published, this will include your full peer review and any attached files.

Reviewer #1: No

Reviewer #2: No

---

## [Author Response · Author response to Decision Letter 0]

26 Jan 2022

Thanks to both the editor and reviewers for taking your extremely valuable time to review our paper. You comments have been constructive and have improved this work significantly.

---

## [Editor Report · Decision Letter 1]

1 Mar 2022

Drug susceptibility profiles and factors associated with non-tuberculous mycobacteria species circulating among patients diagnosed with pulmonary tuberculosis in Tanzania

PONE-D-21-34246R1

Dear Dr. Maya,

We’re pleased to inform you that your manuscript has been judged scientifically suitable for publication and will be formally accepted for publication once it meets all outstanding technical requirements.

Kind regards,

Olivier Neyrolles

Section Editor

PLOS ONE

---

## [Editor Report · Acceptance letter]

7 Mar 2022

PONE-D-21-34246R1 

Drug susceptibility profiles and factors associated with non-tuberculous mycobacteria species circulating among patients diagnosed with pulmonary tuberculosis in Tanzania 

Dear Dr. Maya:

I'm pleased to inform you that your manuscript has been deemed suitable for publication in PLOS ONE. Congratulations! Your manuscript is now with our production department. 

Kind regards, 

on behalf of

Dr. Olivier Neyrolles 

Section Editor

PLOS ONE